# Design, Synthesis and Pharmacological Evaluation of New Quinoline-Based Panx-1 Channel Blockers

**DOI:** 10.3390/ijms24032022

**Published:** 2023-01-19

**Authors:** Letizia Crocetti, Maria Paola Giovannoni, Gabriella Guerrini, Silvia Lamanna, Fabrizio Melani, Gianluca Bartolucci, Marco Pallecchi, Paola Paoli, Martina Lippi, Junjie Wang, Gerhard Dahl

**Affiliations:** 1Neurofarba, Pharmaceutical and Nutraceutical Section, University of Florence, Via Ugo Schiff 6, 50019 Sesto Fiorentino, Italy; 2Department of Industrial Engineering, University of Florence, Via S. Marta 3, 50136 Florence, Italy; 3Department of Physiology and Biophysics, School of Medicine, University of Miami, 1600 N.W. 10th Avenue, Miami, FL 33136, USA

**Keywords:** pannexins, panx-1 blockers, quinolones, quinolines, molecular modeling, chemical stability, X-ray crystallography

## Abstract

Pannexins are an interesting new target in medicinal chemistry, as they are involved in many pathologies such as epilepsy, ischemic stroke, cancer and Parkinson’s disease, as well as in neuropathic pain. They are a family of membrane channel proteins consisting of three members, Panx-1, Panx-2 and Panx-3, and are expressed in vertebrates. In the present study, as a continuation of our research in this field, we report the design, synthesis and pharmacological evaluation of new quinoline-based Panx-1 blockers. The most relevant compounds **6f** and **6g** show an IC_50_ = 3 and 1.5 µM, respectively, and are selective Panx-1 blockers. Finally, chemical stability, molecular modelling and X-ray crystallography studies have been performed providing useful information for the realization of the project.

## 1. Introduction

Pannexins, identified in 2000 [1], are an appealing target in medicinal chemistry because their involvement in several pathologies has been recently demonstrated [2,3]. Pannexins represent a family of membrane channel proteins consisting of three members, Panx-1, Panx-2 and Panx-3, and are expressed in vertebrates [4,5,6]. Panx-1 is ubiquitously distributed and well characterized by cryo-EM studies carried out in 2020 on human (*Homo sapiens*) and frog (*Xenopus tropicalis*) Panx-1 proteins, which revealed a heptameric homomeric structure [7]. The heptamer consists of seven identical subunits positioned around an axis of symmetry through which the channel permeation pathway passes. Each monomer consists of four transmembrane alpha-helix domains, two extracellular loops, an intracellular loop and amino and carboxy terminal portions exposed within the cytoplasm [8]. The Panx-1 channel is generally closed and is highly unlikely to open in the absence of specific stimuli. Reversible opening of the channel occurs through various physiological and pathophysiological stimuli acting on the Panx1 channel itself or indirectly, involving biomolecules such as ATP, glutamate and other receptor ligands. The pathophysiological roles of Pannexins [9] have been clarified through in vitro and in vivo studies by using channel blockers (some compounds are reported in panel A of Figure 1), channel ultra-expression techniques and Panx1 and Panx2 knockout transgenic mouse models [10,11]. 

Recently, our research group has published two papers in the field of Panx-1 channel blockers, in particular indole- and naphthalene-based blockers, for which we identified some important structural requirements for the activity (Figure 1B,C). The most efficacious compounds of both series were also tested in a mouse model of oxaliplatin-induced neuropathic pain, affording interesting results. They showed an anti-hypersensitivity profile and the ability to significantly increase the mouse pain threshold 45 min after the intrathecally administration of the doses of 1 nmol and 3 nmol [12,13].

As a continuation of this research and considering that for the antimalarian mefloquine (Figure 1A), a Panx-1 inhibitor profile has recently been highlighted [14], in the present study, we report the design, synthesis and pharmacological evaluation of new Panx-1 blockers with a quinoline scaffold (compound of type **A** and **B**, Figure 2). To generate group **A** quinolones, those groups and functions found to enhance inhibitory function in our previously published indoles [12] were inserted into the quinolone backbone. On the other hand, the new quinoline derivatives of type **B** show at position 4 of the scaffold an aniline, which is substituted at the para position with sulfonamides or carboxyl groups.

## 2. Results and Discussion

### 2.1. Chemistry

The procedures followed to obtain all new products are shown in Figure 1, Figure 2 and Figure 3, and the structures were confirmed on the basis of analytical and spectral data. The reagents used were synthesized by us (see Appendix A) or are commercially available. Figure 1 depicts the synthetic pathway yielding the final compounds **6a–g**. The quinolinone intermediate **3** [15], which was synthesized according to the procedure reported in the literature starting from 4-nitroaniline **1**, was treated with the appropriate (substituted) arylalkyl halide and commercially available, anhydrous potassium carbonate (K_2_CO_3_) in dry DMF, producing the N-alkylated derivatives **4a**–**g** (**4a** [15]). The reaction of intermediate **3** with (3-chloropropyl)benzene or 1-bromo-5-phenylpentane as the reagent led to a mixture of N- and O-alkylated derivatives (**4c**,**e** and **5c**,**e**, respectively) in a ratio of approximately 4:1. Finally, the intermediates **4a**–**g** underwent basic hydrolysis with sodium hydroxide (NaOH 10%) and EtOH 96% affording the final acid compounds **6a**–**g**. Because the ^1^H-NMR spectra of the couples of the two isomers (**4c**,**e** and **5c**,**e**) were very similar and did not allow a distinction, to assign the correct structure to both isomers, an NMR spectroscopy study was performed. ^13^C-NMR spectra gave the initial information showing two different chemical shift values (77 ppm and 54 ppm) for the methylene of the spacer, which was linked to nitrogen or to the oxygen, values at a later time assigned to O-CH_2_ and N-CH_2_, respectively. The use of 2D-NMR techniques, including Heteronuclear Single Quantum Correlation (HSQC) and Heteronuclear Multiple Bond Correlation (HMBC), provided complementary information and allowed us to establish the exact structure of the isomers. In particular, the HMBC spectra highlighted the presence of the coupling between the methylene protons and the quinolinone C2 carbon for the N-alkylated compounds (**4c** and **4e**), and obviously not for the O-alkylated isomers (**5c** and **5e**). In addition, in the HMBC spectrum of the O-alkylated **5c** and **5e**, it is possible to see the coupling between methylene protons and the C4 carbon of quinoline (see spectra in Appendix A).

The synthesis of 6-aminquinolinones is depicted in Figure 2. The reduction of the nitro to amino group was achieved on intermediate **4a** by catalytic hydrogenation in Parr apparatus (compound **7** [16]); the latter was both hydrolyzed to acid **8** or alkylated at the amino group with benzyl bromide and K_2_CO_3_, thus yielding compound **9**, which, in turn, was transformed into the acid **10**. 

Finally, in Figure 3 is reported the synthesis to obtain the quinoline derivatives **12a**,**b**, **13c**–**g** and **15**. The 4-chloroquinolines **11a**,**b** (**11a** [17], R = H; **11b** [18], R = COOEt)**,** which were treated with the appropriate amines (commercially available or synthesized, as reported in Appendix A) in dry acetonitrile at reflux furnished compounds **12a**,**b** and the ester derivatives **12c**–**g**. Compound **12c** was further reacted with benzyl bromide and K_2_CO_3_, affording **14**; the latter and the ester derivatives **12c**–**g** were subjected to hydrolysis giving the corresponding acids **15** and **13c**–**g,** respectively.

### 2.2. Biological Evaluation

All new products were screened as Panx-1 blockers at the dose of 50 µM using the Voltage Clamp technique on Xenopus Laevis oocytes expressing Panx-1 channels. The membrane potential of the cells was held at -60 mV, and voltage steps to +60 mV were applied to induce membrane currents. To verify that Panx-1 channels carried the currents, 100 µM carbenoxolone (CBX) was applied in each experiment. At this concentration, CBX is known to yield 100% inhibition of Panx-1 activity [19]. If present, leak currents resistant to CBX were subtracted from the total current to assess % inhibition mediated by the tested compounds. The biological results are shown in Table 1, Table 2 and Table 3. According to Table 1 (compounds of structure A), the elongation of the methylene spacer between the N-1 quinoline and the phenyl ring was not favorable for activity, which decreased moving from compound **6a** (n = 1, I = 53%) to **6e** (n = 5, I = 38%). On the other hand, the introduction of an additional acid group, such as a carboxylic group (compound **6g**) or a primary sulphonamide (compound **6f**), in the para position of the phenyl of compound **6a** resulted in an appreciable increase in potency with inhibition values of 89% and 80%, respectively. The reduction of the nitro group of compound **6a** to an amino group (compound **8**) resulted in a similar potency (53% inhibition for **6a** versus 57% for **8**), whereas the presence of a bulky group such as N,N-dibenzylamino (compound **10**) was unfavorable. Table 2 reports compounds of general structure B, which bears at position 4 an aniline substituted in the para position with sulfonamide or carboxylic functions. The activity was found to increase moving from primary sulfonamide **13c** (44% inhibition), through secondary sulfonamide **13d** (66% inhibition) to tertiary sulfonamide **13e**, which showed 88% inhibition. Interestingly, the latter compound contained a portion of the drug Probenecid. On the other hand, the replacement of the above portion with a bulkier group such as in compound **13f** led to a decrease in the activity (I = 32%). On the primary sulphonamide **13c,** some modifications have been performed that yielded different results: the benzylation on the NH at position 4 (compound **15**) resulted in decreased activity and, what is more, the elimination of the carboxylic group at position 3 of the quinolone scaffold (compound **12a**, 10% inhibition). Instead, the replacement of the primary sulphonamide SO_2_NH_2_ with the carboxylic group COOH was highly favorable for activity and compound **13g** achieved 79% inhibition. The elimination of the carboxylic group from position 3 of compound **13g** led to a collapse in activity (compound **12b**, 32% inhibition), which is similar to that observed moving from **13c** to **12a**; this phenomenon confirms that in this position, the presence of an acid group is important.

For the most potent compounds (**6g**, **6f**, **13e** and **13g**) dose–response curves were established (Figure 3). Additionally, selectivity toward purinergic receptors P2X7 and connexins were tested (Table 3). For reference, CBX was applied to verify that the currents were indeed carried by Panx1 channels and did not represent an unspecific leak. With the exception of **13g**, all compounds show an IC_50_ lower than the reference compound CBX (IC_50_ = 10 μM) [20]. Furthermore, the potent Panx-1 blocker **6g**, which almost completely inhibited the channel at 50 µM, exhibited interesting efficacy with IC_50_ = 1.5 µM. Appreciable inhibition values were also obtained for compounds **6f** and **13e** (IC_50_ = 3.0 and 4.2 µM, respectively). Moreover, we tested the selected compounds **6f**, **6g**, **13e** and **13g** on oocytes expressing the P2X7 receptor, because many P2X7 receptor ligands are also able to act as Panx-1 channels blockers. At the same time, Connexins have been chosen as additional target because, together with Pannexins and Innexins, they belong to the superfamily of “gap junction” proteins. The results clearly indicate that compounds **6f**, **6g** and **13g** were selective for the Panx-1 channel, whereas **13e** also blocked P2X7 receptors, although with a lower potency (32% versus 88% for Panx).

For the most potent compounds **6f**, **6g**, **13e** and **13g**, we performed an in silico ADMET evaluation using SwissADME (http://swissadme.ch) (accessed on 22 September 2022) and pkCSM (http://biosig.unimelb.edu.au) (accessed on 22 September 2022). Appendix A depict the physiochemical properties, lipophilicity and water solubility parameters of selected compounds. The BOILED-Egg (Brain Or IntestinaL EstimateD permeation predictive model) plot indicates that our compounds have a low probability of passing passively through the blood–brain barrier because they are out of the yellow area. At the same time, only compound **13g** (molecule 2) has a high probability of being absorbed by the gastrointestinal tract (Figure 4).

Compounds are predicted to be moderately soluble (10^−^^5^/^−^^6^ mol/L range) with low gastrointestinal absorption except for **13g**, which was predicted to be an inhibitor for CYP1A2 but not for the other cytochrome. Compounds **6f**, **6g** and **13e** were predicted inhibitors for CYP2C9; **13e** was predicted to be an inhibitor also for CYP2C19 and CYP3A4. None of the compounds were predicted substrates for these CYP (Appendix A). Our compounds were predicted with low intestinal absorption after oral administration as well as low skin permeability (Log Kp < −2.5), and **6g**, **13e** and **13g** were predicted as PGP substrate (Table 4).

None of the compounds violated the Lipinski rule, but with respect to the other drug-likeness rules, two compounds (**6f** and **6g**) violated Veber, Egan and Muegge rules for their high Topological Polar Surface Area (TPSA) value (Appendix A).

Regarding the distribution properties, the calculated values of steady-state volume of distribution are low (Log VDss < −0.15) indicating that the compounds could be distributed in plasma but to have a significant unbound fraction in the same, thus, to be more available for the target (Table 5).

Finally, the excretion predicted parameters (total clearance) show that compounds could be eliminated by the kidneys (0.34–0.89 mL/min/kg), whereas none of the compounds were a substrate for Organic Cation Transporter 2 (OCT2) (Appendix A).

Because the cLogP values of two potent Panx-1 blockers **6f** and **6g** (1.3036 and 2.3544, respectively, Appendix A) predicted low intestinal absorption by oral administration and the ester precursors of type **4** had low active or inactive results, we found it interesting to evaluate the stability of these latter compounds in human plasma. In fact, if activation by esterases took place in plasma, we could consider ester derivatives **4f** and **4g** to be the pro-drugs of the active acids **6f** and **6g**. Unfortunately, our hypothesis is not supported by stability data, which showed that the ester derivatives are highly stable in human plasma samples with *t*_1/2_ > 120 min. 

### 2.3. Chemical Stability Tests

In general, the presence of an ester group in the structure of a new drug candidate compound leads to the performance of a stability study to verify its possible hydrolysis by esterase enzymes. Thus, the chemical stability of some ester derivatives (**4a**, **4b**, **4d**, **4e** and **4f**) was investigated in phosphate buffer solution (PBS) and in human plasma samples. The stability analyses were performed by high-performance liquid chromatography coupled with a triple quadrupole mass spectrometry system (HPLC-MS/MS) operating in Multiple Reaction Monitoring (MRM) mode. The HPLC-MS/MS instrument and parameters used are reported in Appendix A. 

In these assays, the variation of analytes concentration at four different incubation times both in PBS and in human plasma samples was monitored to evaluate their susceptibility toward spontaneous or enzymatic hydrolysis, respectively. By plotting any variation of analyte concentration vs. the incubation time, the corresponding degradation profiles in tested matrices were obtained. The analyte concentration (1 µM) used during the stability tests is generally smaller than its Michaelis–Menten constant (K_M_), and the enzymatic degradation rate is described by a first-order kinetic. Therefore, by plotting the natural logarithm of the quantitative data versus the incubation time, a linear function can be used, and its slope represents the degradation rate (*k*). Accordingly, with the linear function, the half-life (*t*_1/2_) of each tested compound can be calculated as follows: $$t_{1/2}=\frac{\ln\;(0.50\;\mu M)}{k}$$

The plots of the natural logarithm of the quantitative data versus the incubation time of all the studied compounds were analyzed. For the compounds that showed a value of *k* rate < −0.006 (ln(µM)/min.), a high *t*_1/2_ value should be determined. However, by considering the measure errors and that the highest incubation time involved in the study was 120 min, the *t*_1/2_ parameter of compounds with a low *k* rate will be calculated up to the limit value of 120 min. Parameters beyond this value will be indicated as >120 min.

The obtained degradation profiles showed a remarkably low *k* rate < −0.006 (ln(µM)/min.) for all tested compounds; then, as reported above, they can be considered stable both in PBS and in human plasma samples (*t*_1/2_ > 120 min). The hydrolytic activity of the employed pool of human plasma was checked by adding KEE (reference compound) and, after application of the same procedure followed for the studied compounds, its degradation plot (SFxx) showed a decrease comparable to that found in the literature (*t*_1/2_ ≤ 120 min), confirming the activity of plasma enzymes [21]. The degradation plots in PBS and human plasma of our ester compounds are reported in Appendix A.

### 2.4. X-ray Crystallography

In the asymmetric unit of **4d**, two independent not superimposable (see Figure 5) molecules of **4d** are present. The **4d** compound is, as expected, an N-alkylated one. Both the two independent molecules (A and B) present an almost planar skeleton (defined by the non-hydrogen atoms of the 4-quinolone moiety; the max deviation from the plane is due to C9: 0.032(4) and 0.066(4) Å, for A and B, respectively) that forms angles of 13.3(2)/9.9(1) and 29.9(1)/21.3(2)° with the nitro and the ester moieties (A/B, respectively). In addition, in the A molecule, the carbonyl oxygen atom of the ester group is cis disposed with respect to the oxygen atom of the 4- quinolone moiety, whereas in the B molecule, the two atoms are trans-disposed (see Table 6 and Figure 6). Moreover, the orientation of the two N-alkyl pendants is different. In fact, although in A, the phenyl ring forms an angle of 22.9(1)° with the mean plane containing the 4-quinolone moiety, in B, the angle is 72.3(1)° (see Figure 6).

In addition, each A and B molecule is involved in a short π–π interaction with a symmetry-related molecule (the quinolone rings of A interact with the symmetry-related, 2-x, -y, 1-z rings being the distance between the mean planes 3.38(1) Å, whereas those of B interact with the ones reported by the following symmetry operation: 1-x, 1-y, 1-z, with their distance of 3.27(1) Å). As a consequence, A^…^A and B^…^B dimers are present in the crystal packing (see Figure 7).

### 2.5. Molecular Modeling

To rationalize the results obtained from the in vitro studies, we performed a docking study using the hPANX1 channel–CBX complex, which was resolved by cryo-EM in 2020 [7], with one of the proteins deposited in the Protein Data Bank (6WBI). For molecular modeling studies, we used the Autodock 4.2 program on a reduced hPANX1R, making a cut-off at 2.5 nm (25 Å) from the center of the CBX site. CBX was initially evaluated to confirm the interactions already shown by cryo-EM, as reported in Figure 8. Figure 9, Figure 10 and Figure 11 show the docking of the more potent Panx1 blockers, **6g** (I = 89%) and **13e** (I = 88%), and inactive product **10**. Figure 8 shows the docking of CBX (cluster of five similar conformations within 2 Å) in the channel where interactions with Arg75 of the E-chain and Trp74 of the D-chain can be seen, confirming the importance of these two residues for the binding of CBX to the entrance of the extracellular domain of the channel. The other cluster of CBX conformations keeps the molecule in the center of the channel but oriented toward the Arg75 and Trp74 residues of the other chain of the protein. As can be seen from the figure, CBX is located at the entrance of the channel, and probably, the Trp74 and Arg75 residues of each chain provide the ideal chemical environment to accommodate CBX, which can thus function as a channel blocker.

The docking poses of compound **6g** are illustrated in Figure 9. Of the conformations proposed by AutoDock, 30% show lower free energy (about −10.5 kcal/mol), whereas the remaining others have energy in the range of −8.7 kcal/mol. From the docking analysis, it can be seen that the three lower-energy poses ‘locate’ in different areas of the channel extracellular space (Figure 9A). As can be seen in the figure, **6g** finds the possibility of docking between both chains (the two protomers) C/D and in the channel, analogous to CBX. The possibility of accommodation between the chains could be explained by taking into account the small size of the molecule, which is, therefore, able to move within the extracellular domain. Figure 9B highlights (in green) the strongest interactions between **6g** and the amino acid residues in the three different poses with lower free energy. The amino acids involved are Arg75 and Trp74 of different chains, as shown in the figures.

Similar docking results are observed for compound **13e**, shown in Figure 10. Again, the two lowest-energy poses (−9.0 kcal/mol and −8.1 kcal/mol) find localization between two protomers or in the channel (Figure 10A). On the other hand, **13e** shows another cluster of three higher-energy poses (−6.8 kcal/mol) that are placed in the channel, probably due to the bulkier structure of **13e.** In Figure 10B, the interaction (in green) between **13e** and amino acid residues in the two different accommodations at lower energy, Ile60 and Trp74-Arg75, respectively, are reported. Finally, the docking study are in agreement with the low potency of compound **10** in blocking the channel. In all conformational clusters, **10** is located in the center of the channel, as is CBX, but the distance between ligand and amino acid residues is >0.35 nm; thus, forming hydrogen bonds and/or Van der Walls interactions is (Figure 11) made difficult.

These are only preliminary studies considering the “CBX binding site”. In the future, it will be interesting to further study this site by considering the channel in its entirety to see if there are other more internal binding sites to which our molecules could bind given their smaller size compared to carbenoxolone. 

## 3. Materials and Methods

### 3.1. Chemistry

All melting points were determined on a Büchi apparatus (New Castle, DE, USA) and are uncorrected. Extracts were dried over Na_2_SO_4_, and the solvents were removed under reduced pressure. Merck F-254 commercial plates (Merck, Durham, NC, USA) were used for analytical TLC to follow the course of reactions. Silica gel 60 (Merck 70–230 mesh, Merck, Durham, NC, USA) was used for column chromatography. ^1^H-NMR, ^13^C-NMR, HSQC and HMBC spectra were recorded on an Avance 400 instrument (Bruker Biospin Version 002 with SGU, Bruker Inc., Billerica, MA, USA). Chemical shifts (d) are in parts per million (ppm) approximated by the nearest 0.01 ppm, using the solvent as internal standard. Coupling constants (J) are in Hz; they were calculated by Top Spin 3.1 and approximated by 0.1 Hz. Data are reported as follows: chemical shift, multiplicity (exch—exchange; br—broad; s—singlet; d—doublet; t—triplet; q—quartet; m—multiplet; or a combination of those, e.g.: dd), integral, assignments, and coupling constant. Mass spectra (m/z) were recorded on an ESI-MS triple quadrupole (Varian 1200L) system, in positive ion mode, by infusing a 10 mg/L solution of each analyte dissolved in a mixture of mQ H_2_O:acetonitrile 1:1 *v*/*v*. All new compounds possess a purity ≥ 95%; microanalyses indicated by the symbols of the elements were performed with a Perkin–Elmer 260 elemental analyzer for C, H, and N, and they were within ±0.4% of the theoretical values.

*General procedure for compounds 4b–g and 5c,e*. A mixture of intermediate 3 [15] (0.46 mmol) and anhydrous K_2_CO_3_ (0.92 mmol) in 5 mL of dry DMF was stirred at room temperature for 30′. Then, the appropriate aromatic halide (0.69 mmol) was added, and the mixture was stirred at 80–90 °C for 3 h. After cooling, the mixture was concentrated in vacuo, and ice-cold water (10 mL) was added. Compound **4b** was recovered by vacuum filtration, whereas all others (**4c**–**g** and **5c**,**e**) were recovered by extraction with ethyl acetate (3 × 15 mL). The compound 4b was purified by crystallization from ethanol, whereas **4c**–**g** and **5c**,**e** were purified by flash column chromatography using dichloromethane/methanol 95:5 (for **4c**,**d**,**f** and **5c**) or 99:1 (for **4e** and **5e**) and cyclohexane/ethyl acetate 1:2 (for **4g**) as eluents. Compounds **4c**/**5c** and **4e**/**5e** were obtained as a mixture of isomers of the same reaction.

*Ethyl 6-nitro-4-oxo-1-phenethyl-1,4-dihydroquinoline-3-carboxylate (4b).* Yield = 50%; mp = 128–129 °C (EtOH). ^1^H-NMR (400 MHz, DMSO-d_6_) δ 1.24 (t, 3H, OCH_2_*CH*_3_, *J* = 7.2 Hz), 3.08 (t, 2H, N-CH_2_*CH*_2_-Ph, *J* = 7.2 Hz), 4.18 (q, 2H, O*CH*_2_CH_3_, *J* = 7.2 Hz), 4.67 (t, 2H, N-*CH*_2_CH_2_-Ph, *J* = 7.2 Hz), 7.20–7.28 (m, 5H, Ar), 8.13 (d, 1H, Ar, *J* = 9.2 Hz), 8.46 (s, 1H, Ar), 8.51 (dd, 1H, Ar, *J*_1_ = 2.8 Hz and *J*_2_ = 9.2 Hz), 8.92 (d, 1H, Ar, *J* = 2.8 Hz). ESI-MS calcd. For C_20_H_18_N_2_O_5_, 366.37; found: m/z 367.12 [M + H]^+^. Anal. C_20_H_18_N_2_O_5_ (C, H, N).

*Ethyl 6-nitro-4-oxo-1-(3-phenylpropyl)-1,4-dihydroquinoline-3-carboxylate (4c).* Yield = 32%; oil. ^1^H-NMR (400 MHz, CDCl_3_) δ 1.43 (t, 3H, OCH_2_*CH*_3_, *J* = 7.2 Hz), 2.28 (quin, 2H, N-CH_2_*CH*_2_CH_2_-Ph, *J* = 7.2 Hz), 2.79 (t, 2H, N-CH_2_CH_2_*CH*_2_-Ph, *J* = 7.0 Hz), 4.20 (t, 2H, N-*CH*_2_CH_2_CH_2_-Ph, *J* = 7.4 Hz), 4.40 (q, 2H, O*CH*_2_CH_3_, *J* = 7.2 Hz), 7.21 (d, 2H, Ar, *J* = 6.8 Hz), 7.30–7.37 (m, 4H, Ar), 8.40 (dd, 1H, Ar, *J*_1_ = 2.8 Hz and *J*_2_ = 9.2 Hz), 8.44 (s, 1H, Ar), 9.32 (d, 1H, Ar, *J* = 2.8 Hz). ESI-MS calcd. For C_21_H_20_N_2_O_5_, 380.40; found: m/z 381.14 [M + H]^+^. Anal. C_21_H_20_N_2_O_5_ (C, H, N).

*Ethyl 6-nitro-4-oxo-1-(4-phenylbutyl)-1,4-dihydroquinoline-3-carboxylate (4d).* Yield = 49%; mp = 113–115 °C (EtOH). ^1^H-NMR (400 MHz, CDCl_3_) δ 1.42 (t, 3H, OCH_2_*CH*_3_, *J* = 7.2 Hz), 1.79 (quin, 2H, N-CH_2_CH_2_*CH*_2_CH_2_-Ph, *J* = 7.6 Hz), 1.92 (quin, 2H, N-CH_2_*CH*_2_CH_2_CH_2_-Ph, *J* = 7.6 Hz), 2.71 (t, 2H, N-CH_2_CH_2_CH_2_*CH*_2_-Ph, *J* = 7.2 Hz), 4.20 (t, 2H, N-*CH*_2_CH_2_CH_2_CH_2_-Ph, *J* = 7.6 Hz), 4.39 (q, 2H, O*CH*_2_CH_3_, *J* = 7.2 Hz), 7.15–7.31 (m, 5H, Ar), 7.43 (d, 1H, Ar, *J* = 9.2 Hz), 8.40 (dd, 1H, Ar, *J*_1_ = 2.6 Hz and *J*_2_ = 9.6 Hz), 8.44 (s, 1H, Ar), 9.27 (d, 1H, Ar, *J* = 2.6 Hz). ESI-MS calcd. For C_22_H_22_N_2_O_5_, 394.43; found: m/z 395.16 [M + H]^+^. Anal. C_22_H_22_N_2_O_5_ (C, H, N).

*Ethyl 6-nitro-4-oxo-1-(5-phenylpentyl)-1,4-dihydroquinoline-3-carboxylate (4e).* Yield = 39%; oil. ^1^H-NMR (400 MHz, CDCl_3_) δ 1.42 (t, 3H, OCH_2_*CH*_3_, *J* = 7.2 Hz), 1.45–1.50 (m, 2H, N-CH_2_CH_2_*CH*_2_CH_2_CH_2_-Ph), 1.70 (quin, 2H, N-CH_2_CH_2_CH_2_*CH*_2_CH_2_-Ph, *J* = 7.6 Hz), 1.92 (quin, 2H, N-CH_2_*CH*_2_CH_2_CH_2_CH_2_-Ph, *J* = 7.6 Hz), 2.63 (t, 2H, N-CH_2_CH_2_CH_2_CH_2_*CH*_2_-Ph, *J* = 7.6 Hz), 4.21 (t, 2H, N-*CH*_2_CH_2_CH_2_CH_2_CH_2_-Ph, *J* = 7.6 Hz), 4.39 (q, 2H, O*CH*_2_CH_3_, *J* = 7.2 Hz), 7.12–7.18 (m, 3H, Ar), 7.25–7.30 (m, 2H, Ar), 7.54 (d, 1H, Ar, *J* = 9.2 Hz), 8.42 (dd, 1H, Ar, *J*_1_ = 2.8 Hz and *J*_2_ = 9.2 Hz), 8.45 (s, 1H, Ar), 9.24 (d, 1H, Ar, *J* = 2.8 Hz). ESI-MS calcd. For C_23_H_24_N_2_O_5_, 408.45; found: m/z 409.17 [M + H]^+^. Anal. C_23_H_24_N_2_O_5_ (C, H, N).

*Ethyl 6-nitro-4-oxo-1-(4-sulfamoylbenzyl)-1,4-dihydroquinoline-3-carboxylate (4f).* Yield = 31%; mp = 192–194 °C (EtOH). ^1^H-NMR (400 MHz, DMSO-d_6_) δ 1.30 (t, 3H, OCH_2_*CH*_3_, *J* = 6.8 Hz), 4.27 (q, 2H, O*CH*_2_CH_3_, *J* = 6.8 Hz), 5.84 (s, 2H, N-*CH*_2_-Ph), 7.35 (exch br s, 2H, SO_2_NH_2_), 7.43 (d, 2H, Ar, *J* = 7.2 Hz), 7.77 (t, 3H, Ar, *J* = 8.0 Hz), 8.42 (d, 1H, Ar, *J* = 8.0 Hz), 8.93 (s, 1H, Ar), 9.03 (s, 1H, Ar). ESI-MS calcd. For C_19_H_17_N_3_O_7_S, 431.42; found: m/z 432.08 [M + H]^+^. Anal. C_19_H_17_N_3_O_7_S (C, H, N).

*Ethyl 1-(4-(methoxycarbonyl)benzyl)-6-nitro-4-oxo-1,4-dihydroquinoline-3-carboxylate (4g).* Yield = 29%; mp = 243–245 °C (EtOH). ^1^H-NMR (400 MHz, CDCl_3_) δ 1.43 (t, 3H, OCH_2_*CH*_3_, *J* = 7.0 Hz), 3.92 (s, 3H, OCH_3_), 4.44 (q, 2H, O*CH*_2_CH_3_, *J* = 7.0 Hz), 5.51 (s, 2H, N-*CH*_2_-Ph), 7.24 (d, 2H, Ar, *J* = 8.0 Hz), 7.37 (d, 1H, Ar, *J* = 9.2 Hz), 8.06 (d, 2H, Ar, *J* = 8.0 Hz), 8.34 (dd, 1H, Ar, *J*_1_ = 2.0 Hz and *J*_2_ = 9.2 Hz), 8.65 (s, 1H, Ar), 9.34 (d, 1H, Ar, *J* = 2.4 Hz). ESI-MS calcd. For C_21_H_18_N_2_O_7_, 410.38; found: m/z 411.11 [M + H]^+^. Anal. C_21_H_18_N_2_O_7_ (C, H, N).

*Ethyl 6-nitro-4-(3-phenylpropoxy)quinoline-3-carboxylate (5c).* Yield = 10%; oil. ^1^H-NMR (400 MHz, CDCl_3_) δ 1.42 (t, 3H, OCH_2_*CH*_3_, *J* = 7.0 Hz), 2.31 (quin, 2H, N-CH_2_*CH*_2_CH_2_-Ph, *J* = 7.4 Hz), 2.88 (t, 2H, N-CH_2_CH_2_*CH*_2_-Ph, *J* = 7.4 Hz), 4.38–4.48 (m, 4H, N-*CH*_2_CH_2_CH_2_-Ph + O*CH*_2_CH_3_), 7.20–7.29 (m, 5H, Ar), 8.26 (d, 1H, Ar, *J* = 8.8 Hz), 8.56 (dd, 1H, Ar, *J*_1_ = 2.4 Hz and *J*_2_ = 8.8 Hz), 9.22 (d, 1H, Ar, *J* = 2.4 Hz), 9.24 (s, 1H, Ar). ESI-MS calcd. For C_21_H_20_N_2_O_5_, 380.40; found: m/z 381.14 [M + H]^+^. Anal. C_21_H_20_N_2_O_5_ (C, H, N).

*Ethyl 6-nitro-4-((5-phenylpentyl)oxy)quinoline-3-carboxylate (5e).* Yield = 10%; oil. ^1^H-NMR (400 MHz, CDCl_3_) δ 1.49 (t, 3H, OCH_2_*CH*_3_, *J* = 7.2 Hz), 1.58 (quin, 2H, N-CH_2_CH_2_*CH*_2_CH_2_CH_2_-Ph, *J* = 7.6 Hz), 1.73 (quin, 2H, N-CH_2_CH_2_CH_2_*CH*_2_CH_2_-Ph, *J* = 7.6 Hz), 1.99 (quin, 2H, N-CH_2_*CH*_2_CH_2_CH_2_CH_2_-Ph, *J* = 7.6 Hz), 2.66 (t, 2H, N-CH_2_CH_2_CH_2_CH_2_*CH*_2_-Ph, *J* = 7.6 Hz), 4.40 (t, 2H, N-*CH*_2_CH_2_CH_2_CH_2_CH_2_-Ph, *J* = 6.8 Hz), 4.49 (q, 2H, O*CH*_2_CH_3_, *J* = 7.2 Hz), 7.12–7.18 (m, 3H, Ar), 7.21–7.26 (m, 2H, Ar), 8.36 (d, 1H, Ar, *J* = 9.2 Hz), 8.58 (dd, 1H, Ar, *J*_1_ = 2.4 Hz and *J*_2_ = 9.2 Hz), 9.20 (d, 1H, Ar, *J* = 2.4 Hz), 9.26 (s, 1H, Ar),. ESI-MS calcd. For C_23_H_24_N_2_O_5_, 408.45; found: m/z 409.17 [M + H]^+^. Anal. C_23_H_24_N_2_O_5_ (C, H, N).

*General procedure for compounds 6a–g*. A mixture of suitable ester of type 4 (0.14 mmol), NaOH 10% (1 mL) and EtOH 96% (1 mL) was stirred at reflux for 30 min. After cooling, ice-cold water was added, the mixture was acidified with HCl 6N, and the precipitate was recovered by vacuum filtration and recrystallized with ethanol.

*1-Benzyl-6-nitro-4-oxo-1,4-dihydroquinoline-3-carboxylic acid (6a).* Yield = 66%; mp > 300 °C (EtOH). ^1^H-NMR (400 MHz, DMSO-d_6_) δ 5.92 (s, 2H, N-CH_2_-Ph), 7.30–7.35 (m, 5H, Ar), 8.04 (d, 1H, Ar, *J* = 9.6 Hz), 8.56 (d, 1H, Ar, *J* = 9.2 Hz), 9.02 (s, 1H, Ar), 9.36 (s, 1H, Ar), 14.37 (exch br s, 1H, COOH). ^13^C-NMR (100 MHz, DMSO-d_6_) δ 57.2 (CH_2_), 121.1 (CH), 122.3 (CH), 126.4 (C), 127.1 (CH), 128.1 (CH), 128.6 (CH), 129.5 (CH), 135.5 (C), 143.4 (C), 144.9 (C), 152.4 (CH), 165.6 (C), 177.9 (C). ESI-MS calcd. for C_17_H_12_N_2_O_5_, 324.29; found: m/z 325.08 [M + H]^+^. Anal. C_17_H_12_N_2_O_5_ (C, H, N).

*6-Nitro-4-oxo-1-phenethyl-1,4-dihydroquinoline-3-carboxylic acid (6b)*. Yield = 62%; mp > 300 °C (EtOH). ^1^H-NMR (400 MHz, DMSO-d_6_) δ 3.06 (t, 2H, N-CH_2_CH_2_-Ph, *J* = 7.2 Hz), 4.59 (t, 2H, N-CH_2_CH_2_-Ph, *J* = 7.2 Hz), 7.20–7.30 (m, 5H, Ar), 8.00 (d, 1H, Ar, *J* = 9.2 Hz), 8.40 (d, 1H, Ar, *J* = 7.6 Hz), 8.54 (s, 1H, Ar), 9.00 (s, 1H, Ar). ^13^C-NMR (100 MHz, DMSO-d_6_) δ 34.7 (CH_2_), 59.4 (CH_2_), 107.6 (CH), 109.3 (C), 124.3 (CH), 125.9 (CH), 127.7 (C), 128.6 (CH), 136.8 (C), 138.5 (CH), 139.4 (C), 145.1 (C), 149.1 (CH), 166.2 (C), 176.4 (C). ESI-MS calcd. for C_18_H_14_N_2_O_5_, 338.32; found: m/z 339.09 [M + H]^+^. Anal. C_18_H_14_N_2_O_5_ (C, H, N).

*6-Nitro-4-oxo-1-(3-phenylpropyl)-1,4-dihydroquinoline-3-carboxylic acid (6c).* Yield = 99%; mp > 300 °C (EtOH). ^1^H-NMR (400 MHz, DMSO-d_6_) δ 2.10–2.15 (m, 2H, N-CH_2_CH_2_CH_2_-Ph), 2.71 (t, 2H, N-CH_2_CH_2_CH_2_-Ph, *J* = 7.4 Hz), 4.63 (t, 2H, N-CH_2_CH_2_CH_2_-Ph, *J* = 7.4 Hz), 7.15–7.22 (m, 5H, Ar), 8.22 (d, 1H, Ar, *J* = 9.2 Hz), 8.59 (d, 1H, Ar, *J* = 7.6 Hz), 9.01 (s, 1H, Ar), 9.06 (s, 1H, Ar), 14.37 (exch br s, 1H, COOH). ^13^C-NMR (100 MHz, DMSO-d_6_) δ 27.0 (CH_2_), 31.1 (CH_2_), 48.4 (CH_2_), 107.6 (CH), 109.3 (C), 124.3 (CH), 126.0 (CH), 128.1 (CH), 128.8 (CH), 136.8 (C), 138.5 (CH), 142.0 (C), 145.1 (C), 149.1 (CH), 165.2 (C), 174.4 (C). ESI-MS calcd. for C_19_H_16_N_2_O_5_, 352.35; found: m/z 353.11 [M + H]^+^. Anal. C_19_H_16_N_2_O_5_ (C, H, N).

*6-Nitro-4-oxo-1-(4-phenylbutyl)-1,4-dihydroquinoline-3-carboxylic acid (6d).* Yield = 62%; mp = 175–177 °C (EtOH). ^1^H-NMR (400 MHz, DMSO-d_6_) δ 1.60–1.70 (m, 2H, N-CH_2_CH_2_CH_2_CH_2_-Ph), 1.77–1.85 (m, 2H, N-CH_2_CH_2_CH_2_CH_2_-Ph), 2.61 (t, 2H, N-CH_2_CH_2_CH_2_CH_2_-Ph, *J* = 7.6 Hz), 4.63 (t, 2H, N-CH_2_CH_2_CH_2_CH_2_-Ph, *J* = 7.2 Hz), 7.13–7.27 (m, 5H, Ar), 8.24 (d, 1H, Ar, *J* = 9.6 Hz), 8.61 (dd, 1H, Ar, *J*_1_ = 2.4 Hz and *J*_2_ = 9.2 Hz), 9.02 (d, 1H, Ar, *J* = 2.4 Hz), 9.15 (s, 1H, Ar), 14.45 (exch br s, 1H, COOH). ^13^C-NMR (100 MHz, DMSO-d_6_) δ 28.0 (CH_2_), 28.7 (CH_2_), 35.0 (CH_2_), 54.3 (CH_2_), 109.6 (C), 120.9 (CH), 122.3 (CH), 126.0 (C), 126.3 (CH), 128.0 (CH), 128.1 (CH), 142.1 (C), 143.1 (C), 144.9 (C), 151.9 (CH), 165.6 (C), 177.9 (C). ESI-MS calcd. for C_20_H_18_N_2_O_5_, 366.37; found: m/z 367.12 [M + H]^+^. Anal. C_20_H_18_N_2_O_5_ (C, H, N).

*6-Nitro-4-oxo-1-(5-phenylpentyl)-1,4-dihydroquinoline-3-carboxylic acid (6e).* Yield = 90%; mp = 147–148 °C (EtOH). ^1^H-NMR (400 MHz, DMSO-d_6_) δ 1.30–1.35 (m, 2H, N-CH_2_CH_2_CH_2_CH_2_CH_2_-Ph), 1.55–1.60 (m, 2H, N-CH_2_CH_2_CH_2_CH_2_CH_2_-Ph), 1.75–1.80 (m, 2H, N-CH_2_CH_2_CH_2_CH_2_CH_2_-Ph), 2.50–2.55 (m, 2H, N-CH_2_CH_2_CH_2_CH_2_CH_2_-Ph), 4.45–4.50 (m, 2H, N-CH_2_CH_2_CH_2_CH_2_CH_2_-Ph), 7.12–7.20 (m, 5H, Ar), 8.11 (d, 1H, Ar, *J* = 9.2 Hz), 8.51 (d, 1H, Ar, *J* = 8.8 Hz), 8.89 (s, 1H, Ar), 9.00 (s, 1H, Ar). ^13^C-NMR (100 MHz, DMSO-d_6_) δ 25.8 (CH_2_), 28.9 (CH_2_), 30.9 (CH_2_), 35.4 (CH_2_), 54.2 (CH_2_), 120.0 (C), 122.3 (CH), 126.1 (CH), 127.8 (CH), 128.7 (CH), 142.4 (C), 143.1 (C), 144.0 (C), 151.0 (CH), 165.4 (C), 166.9 (C), 177.0 (C). ESI-MS calcd. for C_21_H_20_N_2_O_5_, 408.45; found: m/z 409.17 [M + H]^+^. Anal. C_23_H_24_N_2_O_5_ (C, H, N).

*6-Nitro-4-oxo-1-(4-sulfamoylbenzyl)-1,4-dihydroquinoline-3-carboxylic acid (6f).* Yield = 31%; mp > 300 °C (EtOH). ^1^H-NMR (400 MHz, DMSO-d_6_) δ 6.00 (s, 2H, N-CH_2_-Ph), 7.35 (exch br s, 2H, SO_2_NH_2_), 7.46 (d, 2H, Ar, *J* = 8.4 Hz), 7.77 (d, 2H, Ar, *J* = 8.4 Hz), 7.95 (d, 1H, Ar, *J* = 9.2 Hz), 8.55 (dd, 1H, Ar, *J*_1_ = 2.4 Hz and *J*_2_ = 9.2 Hz), 9.03 (d, 1H, Ar, *J* = 2.8 Hz), 9.39 (s, 1H, Ar), 14.25 (exch br s, 1H, COOH). ^13^C-NMR (100 MHz, DMSO-d_6_) δ 55.8 (CH_2_), 126.1 (CH), 126.7 (CH), 127.4 (CH), 128.3 (CH), 138.5 (CH), 140.0 (C), 143.3 (C), 144.1 (CH), 150.0 (C), 166.2 (C), 176.4 (C). ESI-MS calcd. for C_17_H_13_N_3_O_7_S, 403.37; found: m/z 404.05 [M + H]^+^. Anal. C_17_H_13_N_3_O_7_S (C, H, N).

*1-(4-Carboxybenzyl)-6-nitro-4-oxo-1,4-dihydroquinoline-3-carboxylic acid (6g).* Yield = 54%; mp > 300 °C (EtOH). ^1^H-NMR (400 MHz, DMSO-d_6_) δ 6.01 (s, 2H, N-CH_2_-Ph), 7.39 (d, 2H, Ar, *J* = 8.4 Hz), 7.89 (d, 2H, Ar, *J* = 8.0 Hz), 7.96 (d, 1H, Ar, *J* = 9.6 Hz), 8.54 (dd, 1H, Ar, *J*_1_ = 2.6 Hz and *J*_2_ = 9.4 Hz), 9.02 (d, 1H, Ar, *J* = 2.4 Hz), 9.39 (s, 1H, Ar), 12.99 (exch br s, 1H, COOH), 14.35 (exch br s, 1H, COOH). ^13^C-NMR (100 MHz, DMSO-d_6_) δ 57.0 (CH_2_), 110.1 (C), 121.1 (CH), 126.4 (C), 127.2 (CH), 127.5 (CH), 128.3 (CH), 129.6 (CH), 130.4 (CH), 131.01 (C), 140.3 (C), 143.3 (C), 145.0 (C), 152.7 (CH), 165.5 (C), 167.3 (C), 178.1 (C). ESI-MS calcd. for C_18_H_12_N_2_O_7_, 368.30; found: m/z 369.07 [M + H]^+^. Anal. C_18_H_12_N_2_O_7_ (C, H, N).

*General procedure for compounds 8 and 10.* Compounds **8** and **10** were obtained following the same procedure performed for compounds **6a–g** but starting from intermediates 7 [16] and 9, respectively. After cooling, ice-cold water (10 mL) was added, and compounds **8** and **10** were recovered by vacuum filtration to obtain a solid which was purified by crystallization with ethanol.

*6-Amino-1-benzyl-4-oxo-1,4-dihydroquinoline-3-carboxylic acid (8).* Yield = 85%; mp > 300 °C (EtOH). ^1^H-NMR (400 MHz, DMSO-d_6_) δ 5.72 (s, 4H, N-CH_2_-Ph + NH_2_), 7.05–7.54 (m, 8H, Ar), 8.96 (s, 1H, Ar), 15.69 (exch br s, 1H, COOH). ^13^C-NMR (100 MHz, DMSO-d_6_) δ 55.8 (CH_2_), 109.3 (C), 114.8 (CH), 122.9 (CH), 126.9 (CH), 127.9 (C), 128.5 (CH), 133.9 (C), 136.1 (C), 137.3 (CH), 148.0 (CH), 166.2 (C), 176.4 (C). ESI-MS calcd. for C_17_H_14_N_2_O_3_, 294.31; found: m/z 295.10 [M + H]^+^. Anal. C_17_H_14_N_2_O_3_ (C, H, N).

*1-Benzyl-6-(dibenzylamino)-4-oxo-1,4-dihydroquinoline-3-carboxylic acid (10).* Yield = 86%; mp = 215–217 °C (EtOH). ^1^H-NMR (400 MHz, DMSO-d_6_) δ 4.84 (s, 4H, 2 x N-CH_2_-Ph), 5.74 (s, 2H, N-CH_2_-Ph), 7.21–7.33 (m, 16H, Ar), 7.38 (d, 1H, Ar, *J* = 2.4 Hz), 7.64 (d, 1H, Ar, *J* = 9.2 Hz), 9.03 (s, 1H, Ar), 15.43 (exch br s, 1H, COOH). ^13^C-NMR (100 MHz, DMSO-d_6_) δ 54.8 (CH_2_), 56.8 (CH_2_), 104.9 (CH), 120.1 (CH), 120.6 (CH), 126.9 (CH), 127.1 (CH), 127.4 (CH), 127.8 (C), 128.5 (CH), 129.1 (CH), 129.4 (CH), 131.2 (C), 136.2 (C), 138.5 (C), 146.8 (CH), 147.3 (CH), 167.2 (C), 177.3 (C). ESI-MS calcd. for C_31_H_26_N_2_O_3_, 474.56; found: m/z 475.20 [M + H]^+^. Anal. C_31_H_26_N_2_O_3_ (C, H, N).

*Ethyl 1-benzyl-6-(dibenzylamino)-4-oxo-1,4-dihydroquinoline-3-carboxylate (9).* Compound **9** was obtained following the same procedure performed for compounds **4a**–**g** but starting from intermediate 7 [16] and using an excess of benzyl bromide as the reagent (3 equ.). After cooling, ice-cold water (10 mL) was added, and the compound was recovered by vacuum filtration to obtain a solid which was purified by crystallization with ethanol. Yield = 90%; mp = 95–97 °C (EtOH). ^1^H-NMR (400 MHz, DMSO-d_6_) δ 1.25 (t, 3H, OCH_2_CH_3_, *J* = 6.8 Hz), 4.18 (q, 2H, OCH_2_CH_3_, *J* = 6.8 Hz), 4.77 (s, 4H, 2 x N-CH_2_-Ph), 5.55 (s, 2H, N-CH_2_-Ph), 7.06 (d, 1H, Ar, *J* = 8.4 Hz), 7.20–7.42 (m, 17H, Ar), 8.71 (s, 1H, Ar). ESI-MS calcd. for C_33_H_30_N_2_O_3_, 502.61; found: m/z 503.23 [M + H]^+^. Anal. C_33_H_30_N_2_O_3_ (C, H, N).

*General procedure for compounds 12a,b.* To a solution of intermediate 11a [17] (0.43 mmol) in 4 mL of anhydrous acetonitrile, 0.47 mmol of 4-aminobenzenesulfonamide (for **12a**) or 4-aminobenzoic acid (for **12b**), which is commercially available, and 1.08 mmol of triethylamine were added. The mixture was stirred at reflux for 16 h. After cooling, the precipitate formed was recovered by vacuum filtration and washed with reaction solvent. The crude product was purified by crystallization with ethanol to obtain the final compound. 

*4-(Quinolin-4-ylamino)benzenesulfonamide (12a).* Yield = 14%; mp > 300 °C (EtOH). ^1^H-NMR (400 MHz, DMSO-d_6_) δ 7.00 (d, 1H, Ar, *J* = 6.4 Hz), 7.48 (exch br s, 2H, SO_2_NH_2_), 7.69 (d, 2H, Ar, *J* = 8.4 Hz), 7.84 (t, 1H, Ar, *J* = 7.6 Hz), 7.98 (d, 2H, Ar, *J* = 8.4 Hz), 8.03–8.10 (m, 2H, Ar), 8.60 (s, 1H, Ar), 8.82 (d, 1H, Ar, *J* = 8.4 Hz), 11.10 (exch br s, 1H, NH). ^13^C-NMR (100 MHz, DMSO-d_6_) δ 112.8 (CH), 113.8 (CH), 121.6 (C), 124.2 (CH), 125.7 (CH), 129.2 (CH), 129.6 (CH), 130.0 (CH), 130.9 (C), 138.7 (C), 149.1 (C), 149.7 (C), 151.1 (CH). ESI-MS calcd. for C_15_H_13_N_3_O_2_S, 299.35; found: m/z 300.08 [M + H]^+^. Anal. C_15_H_13_N_3_O_2_S (C, H, N).

*4-(Quinolin-4-ylamino)benzoic acid (12b).* Yield = 19%; mp > 300 °C (EtOH). ^1^H-NMR (400 MHz, DMSO-d_6_) δ 7.24 (d, 1H, Ar, *J* = 3.2 Hz), 7.41 (d, 2H, Ar, *J* = 8.4 Hz), 7.58 (t, 1H, Ar, *J* = 7.0 Hz), 7.72 (t, 1H, Ar, *J* = 7.2 Hz), 7.92 (d, 3H, Ar, *J* = 8.4 Hz), 8.33 (d, 1H, Ar, *J* = 8.4 Hz), 8.58 (s, 1H, Ar), 9.27 (exch br s, 1H, NH), 12.54 (exch br s, 1H, COOH). ^13^C-NMR (100 MHz, DMSO-d_6_) δ 112.8 (CH), 119.5 (CH), 120.2 (C), 121.6 (C), 124.2 (CH), 125.7 (CH), 129.2 (CH), 129.6 (CH), 131.1 (CH), 138.7 (C), 149.7 (C), 151.1 (C), 151.6 (CH), 169.3 (C). ESI-MS calcd. for C_16_H_12_N_2_O_2_, 264.28; found: m/z 265.09 [M + H]^+^. Anal. C_16_H_12_N_2_O_2_ (C, H, N).

*General procedure for compounds 12c–g.* An amount of 0.93 mmol of appropriate aniline was added to a solution of intermediate **11b** [18] (0.85 mmol) in 4 mL of anhydrous acetonitrile. The mixture was stirred at reflux for 2–3 h. After cooling, the precipitate formed was recovered by vacuum filtration and washed with reaction solvent. The crude product was purified by crystallization with ethanol to obtain the final compound. Only compounds **12d** and **12e** were purified by flash column chromatography using cyclohexane/ethyl acetate 1:3 (for **12d**) and 2:1 (for **12e**) as eluent.

*Ethyl 4-[(4-sulfamoylphenyl)amino]quinoline-3-carboxylate (12c).* Yield = 83%; mp = 227–230 °C (EtOH). ^1^H-NMR (400 MHz, DMSO-d_6_) δ 1.08 (t, 3H, OCH_2_CH_3_, *J* = 7.0 Hz), 3.79 (q, 2H, OCH_2_CH_3_, *J* = 7.0 Hz), 7.40 (exch br s, 2H, SO_2_NH_2_), 7.44 (d, 2H, Ar, *J* = 8.8 Hz), 7.77–7.83 (m, 3H, Ar), 8.06 (t, 1H, Ar, *J* = 7.6 Hz), 8.15 (d, 1H, Ar, *J* = 8.4 Hz), 8.68 (d, 1H, Ar, *J* = 8.4 Hz), 9.05 (s, 1H, Ar), 11.25 (exch br s, 1H, NH). ESI-MS calcd. for C_18_H_17_N_3_O_4_S, 371.41; found: m/z 372.10 [M + H]^+^. Anal. C_18_H_17_N_3_O_4_S (C, H, N).

*Ethyl 4-[4-(N-propylsulfamoyl)phenyl]aminoquinoline-3-carboxylate (12d).* Yield = 13%; oil. ^1^H-NMR (400 MHz, DMSO-d_6_) δ 0.79 (t, 3H, N-CH_2_CH_2_CH_3_, *J* = 7.2 Hz), 1.09 (t, 3H, OCH_2_CH_3_, *J* = 6.8 Hz), 1.34–1.39 (m, 2H, N-CH_2_CH_2_CH_3_), 2.64 (t, 2H, N-CH_2_CH_2_CH_3_, *J* = 6.8 Hz), 3.98 (q, 2H, OCH_2_CH_3_, *J* = 6.8 Hz), 7.08 (d, 2H, Ar, *J* = 7.6 Hz), 7.39 (exch br d, 1H, SO_2_NH-CH_2_, *J* = 5.6 Hz), 7.58–7.64 (m, 3H, Ar), 7.83 (t, 1H, Ar, *J* = 7.0 Hz), 8.02 (d, 1H, Ar, *J* = 8.0 Hz), 8.14 (d, 1H, Ar, *J* = 8.0 Hz), 8.98 (s, 1H, Ar), 9.72 (exch br s, 1H, NH). ESI-MS calcd. for C_21_H_23_N_3_O_4_S, 413.49; found: m/z 414.14 [M + H]^+^. Anal. C_21_H_23_N_3_O_4_S (C, H, N).

*Ethyl 4-[4-(N,N-dipropylsulfamoyl)phenyl]aminoquinoline-3-carboxylate (12e).* Yield = 19%; oil. ^1^H-NMR (400 MHz, CDCl_3_) δ 0.90 (t, 6H, 2 x N-CH_2_CH_2_CH_3_, *J* = 7.6 Hz), 1.52 (t, 3H, OCH_2_CH_3_, *J* = 7.2 Hz), 1.59 (quin, 4H, 2 x N-CH_2_CH_2_CH_3_, *J* = 7.6 Hz), 3.12 (t, 4H, 2 x N-CH_2_CH_2_CH_3_, *J* = 7.6 Hz), 4.48 (q, 2H, OCH_2_CH_3_, *J* = 7.2 Hz), 7.19 (d, 2H, Ar, *J* = 8.0 Hz), 7.31 (t, 1H, Ar, *J* = 7.6 Hz), 7.58 (d, 1H, Ar, *J* = 8.8 Hz), 7.74–7.81 (m, 4H, Ar), 9.33 (s, 1H, Ar), 11.17 (exch br s, 1H, NH). ESI-MS calcd. for C_21_H_23_N_3_O_4_S, 413.49; found: m/z 414.14 [M + H]^+^. Anal. C_21_H_23_N_3_O_4_S (C, H, N).

*Ethyl 4-[4-(N,N-dibenzylsulfamoyl)phenyl]aminoquinoline-3-carboxylate (12f).* Yield = 64%; mp = 257–259 °C (EtOH). ^1^H-NMR (400 MHz, CDCl_3_) 1.49 (t, 3H, OCH_2_CH_3_, *J* = 6.8 Hz), 4.40 (s, 4H, 2 x N-CH_2_-Ph), 4.53 (q, 2H, OCH_2_CH_3_, *J* = 6.8 Hz), 7.10–7.15 (m, 4H, Ar), 7.25–7.35 (m, 9H, Ar), 7.54 (d, 1H, Ar, *J* = 8.4 Hz), 7.89 (d, 3H, Ar, *J* = 7.6 Hz), 8.68 (s, 1H, Ar), 9.35 (s, 1H, Ar), 12.02 (exch br s, 1H, NH). ESI-MS calcd. for C_32_H_29_N_3_O_4_S, 551.66; found: m/z 552.19 [M + H]^+^. Anal. C_32_H_29_N_3_O_4_S (C, H, N).

*4-[3-(Ethoxycarbonyl)quinolin-4-yl]aminobenzoic acid (12g).* Yield = 87%; mp = 247–250 °C (EtOH). ^1^H-NMR (400 MHz, DMSO-d_6_) δ 1.07 (t, 3H, OCH_2_CH_3_, *J* = 7.2 Hz), 3.79 (q, 2H, OCH_2_CH_3_, *J* = 7.2 Hz), 7.38 (d, 2H, Ar, *J* = 8.4 Hz), 7.78 (t, 1H, Ar, *J* = 7.8 Hz), 7.95 (d, 2H, Ar, *J* = 8.4 Hz), 8.06 (t, 1H, Ar, *J* = 7.6 Hz), 8.15 (d, 1H, Ar, *J* = 8.4 Hz), 8.65 (d, 1H, Ar, *J* = 8.4 Hz), 9.03 (s, 1H, Ar), 11.27 (exch br s, 1H, NH). ESI-MS calcd. for C_19_H_16_N_2_O_4_, 336.35; found: m/z 377.11 [M + H]^+^. Anal. C_19_H_16_N_2_O_4_ (C, H, N).

*Ethyl 4-[benzyl(4-sulfamoylphenyl)amino]quinoline-3-carboxylate (14).* Compound **14** was obtained following the same procedure performed for compounds **4a**–**g** and **9** but starting from intermediate **12c** and using an excess of benzyl bromide as the reagent (3 equ.). After cooling, ice-cold water (10 mL) was added, and compound **14** was recovered by vacuum filtration to obtain a solid which was purified by crystallization with ethanol. Yield = 27%; mp = 191 °C dec. (EtOH). ^1^H-NMR (400 MHz, DMSO-d_6_) δ 0.91 (t, 3H, OCH_2_CH_3_, *J = 6*.8 Hz), 3.42 (q, 2H, OCH_2_CH_3_, *J* = 6.8 Hz), 5.52 (s, 2H, N-CH_2_-Ph), 6.88 (d, 2H, Ar, *J* = 7.6 Hz), 7.16 (exch br s, 2H, SO_2_NH_2_), 7.30–7.40 (m, 6H, Ar), 7.44 (d, 1H, Ar, *J* = 8.4 Hz), 7.54 (t, 1H, Ar, *J* = 7.0 Hz), 7.65 (d, 2H, Ar, *J* = 8.0 Hz), 8.32 (s, 1H, Ar), 8.46 (d, 1H, Ar, *J* = 8.0 Hz). ESI-MS calcd. for C_25_H_23_N_3_O_4_S, 461.54; found: m/z 462.14 [M + H]^+^. Anal. C_25_H_23_N_3_O_4_S (C, H, N).

*General procedure for compounds 13c–g and 15.* Compounds **13c**–**g** and **15** were obtained following the same procedure performed for compounds **6a**–**g**, **8** and **10** but starting from intermediates **12c**–**g** and **14**, respectively. After cooling, ice-cold water (10 mL) was added, and compounds **13c**–**g** and **15** were recovered by vacuum filtration to obtain a solid which was purified by crystallization with ethanol.

*4-[(4-Sulfamoylphenyl)amino]quinoline-3-carboxylic acid (13c).* Yield = 97%; mp > 300 °C (EtOH). ^1^H-NMR (400 MHz, DMSO-d_6_) δ 6.88 (d, 2H, Ar, *J* = 8.4 Hz), 7.16 (exch br s, 2H, SO_2_NH_2_), 7.31 (t, 1H, Ar, *J* = 7.2 Hz), 7.55 (d, 1H, Ar, *J* = 8.0 Hz), 7.60–7.65 (m, 3H, Ar), 7.91 (d, 1H, Ar, *J* = 8.4 Hz), 9.27 (s, 1H, Ar), 13.01 (exch br s, 1H, COOH). ^13^C-NMR (100 MHz, DMSO-d_6_) δ 99.9 (CH), 118.6 (C), 118.9 (CH), 121.2 (C), 124.8 (CH), 125.8 (CH), 127.4 (CH), 129.7 (CH), 130.1 (CH), 131.1 (C), 136.8 (C), 148.2 (C), 148.5 (C), 153.8 (CH), 169.3 (C). ESI-MS calcd. for C_16_H_13_N_3_O_4_S, 343.36; found: m/z 344.07 [M + H]^+^. Anal. C_16_H_13_N_3_O_4_S (C, H, N).

*4-[4-(N-Propylsulfamoyl)phenyl]aminoquinoline-3-carboxylic acid (13d).* Yield = 87%; mp > 300 °C (EtOH). ^1^H-NMR (400 MHz, DMSO-d_6_) δ 0.75 (t, 3H, N-CH_2_CH_2_CH_3_, *J* = 6.8 Hz), 1.32 (quin, 2H, N-CH_2_CH_2_CH_3_, *J* = 6.8 Hz), 2.64 (t, 2H, N-CH_2_CH_2_CH_3_, *J* = 6.8 Hz), 6.96 (d, 2H, Ar, *J* = 7.6 Hz), 7.34 (exch br d, 2H, SO_2_NH-CH_2_ + Ar, *J* = 7.2 Hz), 7.58–7.65 (m, 3H, Ar), 7.68 (t, 1H, Ar, *J* = 7.0 Hz), 7.92 (d, 1H, Ar, *J* = 8.0 Hz), 9.23 (s, 1H, Ar), 12.78 (exch br s, 1H, COOH). ^13^C-NMR (100 MHz, DMSO-d_6_) δ 11.6 (CH_3_), 22.8 (CH_2_), 44.8 (CH_2_), 99.9 (CH), 119.3 (CH), 120.5 (C), 125.1 (CH), 125.9 (CH), 128.3 (CH), 129.6 (CH), 130.2 (CH), 133.2 (C), 148.4 (C), 148.9 (C), 149.5 (C), 169.3 (C). ESI-MS calcd. for C_19_H_19_N_3_O_4_S, 385.44; found: m/z 386.11 [M + H]^+^. Anal. C_19_H_19_N_3_O_4_S (C, H, N).

*4-[4-(N,N-Dipropylsulfamoyl)phenyl]aminoquinoline-3-carboxylic acid (13e).* Yield = 37%; mp > 300 °C (EtOH). ^1^H-NMR (400 MHz, DMSO-d_6_) δ 0.80 (t, 6H, 2 x N-CH_2_CH_2_CH_3_, *J* = 6.8 Hz), 1.40–1.50 (m, 4H, 2 x N-CH_2_CH_2_CH_3_), 3.00 (t, 4H, 2 x N-CH_2_CH_2_CH_3_, *J* = 6.8 Hz), 7.14 (d, 2H, Ar, *J* = 7.6 Hz), 7.40–7.45 (m, 1H, Ar), 7.65 (d, 2H, Ar, *J* = 7.2 Hz), 7.70–7.80 (m, 2H, Ar), 7.98 (d, 1H, Ar, *J* = 6.8 Hz), 9.11 (s, 1H, Ar), 13.55 (exch br s, 1H, COOH). ^13^C-NMR (100 MHz, DMSO-d_6_) δ 11.5 (CH_3_), 20.0 (CH_2_), 53.0 (CH_2_), 113.0 (C), 113.8 (C), 120.3 (C), 126.6 (CH), 129.7 (CH), 130.0 (CH), 132.0 (CH), 140.2 (C), 149.9 (C), 151.7 (CH), 153.3 (CH), 168.5 (C), 169.8 (C). ESI-MS calcd. for C_22_H_25_N_3_O_4_S, 427.52; found: m/z 428.16 [M + H]^+^. Anal. C_22_H_25_N_3_O_4_S (C, H, N).

*4-[4-(N,N-Dibenzylsulfamoyl)phenyl]aminoquinoline-3-carboxylic acid (13f).* Yield = 66%; mp > 300 °C (EtOH). ^1^H-NMR (400 MHz, DMSO-d_6_) 4.27 (s, 4H, 2 x N-CH_2_-Ph), 7.08–7.20 (m, 12H, Ar), 7.45–7.50 (m, 1H, Ar), 7.74 (d, 2H, Ar, *J* = 7.6 Hz), 7.82 (d, 2H, Ar, *J* = 7.2 Hz), 8.01 (d, 1H, Ar, *J* = 7.6 Hz), 9.14 (s, 1H, Ar). ^13^C-NMR (100 MHz, DMSO-d_6_) δ 51.5 (CH_2_), 119.8 (CH), 125.8 (CH), 127.9 (CH), 128.7 (CH), 129.1 (CH), 132.4 (CH), 133.2 (C), 136.7 (C), 148.1 (C), 168.2 (C). ESI-MS calcd. for C_30_H_25_N_3_O_4_S, 523.61; found: m/z 524.16 [M + H]^+^. Anal. C_30_H_25_N_3_O_4_S (C, H, N).

*4-[(4-Carboxyphenyl)amino]quinoline-3-carboxylic acid (13g).* Yield = 82%; mp = 259–260 °C (EtOH). ^1^H-NMR (400 MHz, DMSO-d_6_) δ 6.90 (d, 2H, Ar, *J* = 8.0 Hz), 7.34 (t, 1H, Ar, *J* = 7.2 Hz), 7.62 (d, 1H, Ar, *J* = 8.0 Hz), 7.67 (t, 1H, Ar, *J* = 7.6 Hz), 7.78 (d, 2H, Ar, *J* = 8.0 Hz), 7.93 (d, 1H, Ar, *J* = 8.0 Hz), 9.22 (s, 1H, Ar), 12.61 (exch br s, 1H, NH). ^13^C-NMR (100 MHz, DMSO-d_6_) δ 116.4 (C), 119.4 (CH), 120.8 (C), 124.6 (C), 125.2 (CH), 126.1 (CH), 129.0 (CH), 130.7 (CH), 132.1 (CH), 148.6 (C), 149.6 (C), 152.2 (CH), 167.5 (C), 169.9 (C). ESI-MS calcd. for C_17_H_12_N_2_O_4_, 308.29; found: m/z 309.08 [M + H]^+^. Anal. C_17_H_12_N_2_O_4_ (C, H, N).

*4-[Benzyl(4-sulfamoylphenyl)amino]quinoline-3-carboxylic acid (15).* Yield = 71%; mp = 264 °C dec. (EtOH). ^1^H-NMR (400 MHz, DMSO-d_6_) δ 5.92 (s, 2H, N-CH_2_-Ph), 7.40–7.50 (m, 10H, 8H Ar + 2H, SO_2_NH_2_), 7.64 (d, 1H, Ar, *J* = 6.4 Hz), 7.78 (t, 1H, Ar, *J* = 7.6 Hz), 7.82 (d, 2H, Ar, *J* = 8.4 Hz), 7.93 (d, 1H, Ar, *J* = 8.4 Hz), 9.32 (s, 1H, Ar). ^13^C-NMR (100 MHz, DMSO-d_6_) δ 57.1 (CH_2_), 113.8 (C), 113.9 (CH), 119.0 (CH), 120.0 (C), 123.2 (CH), 125.5 (CH), 126.7 (CH), 127.2 (CH), 127.9 (CH), 128.5 (CH), 129.0 (C), 129.4 (CH), 130.0 (CH), 132.0 (CH), 134.1 (CH), 135.9 (C), 139.8 (C), 140.0 (C), 148.9 (CH), 152.6 (C), 166.9 (C). ESI-MS calcd. for C_23_H_19_N_3_O_4_S, 433.48; found: m/z 434.11 [M + H]^+^. Anal. C_23_H_19_N_3_O_4_S (C, H, N).

### 3.2. Electrophysiological Assay

#### 3.2.1. Preparation of Oocytes

Oocytes of *Xenopus laevis* are prepared as previously described [22]. The oocytes are isolated from surgically removed ovary segments incubated in an Oocyte Ringe solution Ca_2_-free (OR2: in mM: 82.5 NaCl, 2.5 KCl, 1.0 MgCl_2_, 1.0 CaCl_2_, 1.0 Na_2_HPO_4_ e 5.0 HEPES, pH 7.5) with 2 mg/mL of collagenase and antibiotics (10,000 U/mL penicillin and 10 mg/mL of streptomycin) stirred at room temperature for 3 h. After being washed with OR2, oocytes free of follicular cells and having a uniform pigmentation are chosen and stored in OR2 for 18 °C.

#### 3.2.2. Synthesis of mRNA

Mouse pannexin1, in pCS2, was linearized with NotI. In vitro transcription was performed with the polymerase SP6, using the Message Machine kit (Ambion, Berlin, Germany). mRNA was quantified by absorbance (260 nm), and the proportion of full-length transcripts was checked by agarose gel electrophoresis. In Vitro-transcribed mRNA (~20 nL) was injected into *Xenopus laevis* oocytes.

#### 3.2.3. Electrophysiology

Whole cell membrane currents of oocytes were measured using a two-electrode voltage clamp (Gene Clamp 500B, Axon Instruments/Molecular Devices Sunnyvale, CA, USA). Glass pipettes were pulled using a P-97 Flaming/Brown type puller (Sutter, Novato, CA, USA). The recording chamber was perfused continuously with frog Ringer (OR) solution (in mM: 82.5 NaCl, 2.5 KCl, 1 CaCl2, 1 MgCl2, 1 Na2HPO4, and 5 HEPES, pH 7.5). Membrane conductance was determined using voltage pulses, and the pulse-induced current amplitudes were divided by the amplitudes of the voltage steps. For calculation of % inhibition, leak currents were subtracted. Typically, leak currents after the interventions were smaller than at the beginning of the experiment, and this smaller value was used for the subtraction to avoid overestimation of the inhibitory effect. For an independent measure of the leak current, oocytes at the end of the experiment were exposed to 100 µM carbenoxolone, which is known to close Panx1 channels 100%. Oocytes expressing Panx1 were held at −60 mV, and pulses to +60 mV were applied to transiently open the channels by means of the voltage gate. Pulses 5 s in duration were applied at 0.1 Hz for current and conductance measurements.

### 3.3. Chemical Stability Data

#### 3.3.1. Instrumental

The HPLC-MS/MS analysis was carried out using a Varian 1200L triple quadrupole system (Palo Alto, CA, USA) equipped by two Prostar 210 pumps, a Prostar 410 autosampler and an Electrospray Source (ESI) operating in positive ions mode. Raw-data were collected and processed by Varian Workstation Vers. 6.8 software. A G-Therm 015 thermostatic oven was used to keep the samples at 37 °C during the degradation tests. Eppendorf microcentrifuge 5415D was employed to centrifuge plasma samples.

#### 3.3.2. Standard Solutions and Calibration Curves

Stock solutions of analytes and verapamil hydrochloride (IS) were prepared in acetonitrile at 1.0 mg mL^−1^ and stored at 4 °C. Working solutions of each analyte were freshly prepared by diluting stock solutions up to a concentration of 10 μM and 1 μM (working solutions 1 and 2, respectively) in mQ water: acetonitrile 80:20 (*v*/*v*) solution. The IS working solution was prepared in acetonitrile at 30 ng mL^−1^ (IS solution). A seven levels calibration curve was prepared by adding proper volumes of working solution of each analyte to 500 μL of IS solution. The obtained solutions were dried under a gentle nitrogen stream and dissolved in 1.0 mL of 10 mM of formic acid in mQ water: acetonitrile 80:20 (*v*/*v*) solution. Final concentrations of calibration levels were: 0, 2.5, 5, 10, 25, 50 and 100 ng mL^−1^ of analyte in the sample. All calibration levels were analyzed by the appropriate LC-MS/MS method.

#### 3.3.3. HPLC-MS/MS Method

The chromatographic parameters were reported as follows:-analytical column, Luna C18 length = 20 mm; internal diameter = 2 mm; particle size = 3 μm -acidic mobile phase, composed by 5 mM of ammonium formate and 10 mM of formic acid in mQ water: acetonitrile 90:10 (*v*/*v*) solution (solvent A), 5 mM of ammonium formate and 10 mM of formic acid in mQ water: acetonitrile 10:90 (*v*/*v*) solution (solvent B). -the injection volume was 5 μL.

The elution gradient is shown in Appendix A. The analyses were acquired in MRM (Multiple Reaction Monitoring), using argon as collision gas, and parameters are reported in Appendix A.

#### 3.3.4. Linearity and LOD

Calibration curves of analytes were obtained by plotting the peak area ratios (PAR), between quantitation ions of analyte and IS, versus the nominal concentration of the calibration solution. A linear regression analysis was applied to obtain the best fitting function between the calibration points. The precision was evaluated through the relative standard deviation (RSD%) of the quantitative data of the replicate analysis of the highest level of calibration curves. In order to obtain reliable LOD values, the standard deviation of response and slope approach was employed. The estimated standard deviations of responses were obtained by the standard deviation of y-intercepts (SDY-I) of regression lines. The obtained linear regressions, the linearity coefficients, the precision and the estimated LOD values for each analyte are reported in Appendix A.

### 3.4. X-ray Crystallography

#### Single-Crystal X-ray Diffraction

Several crystals of 4d were tested by single crystal X-ray diffraction (SCXRD). Unfortunately, none of them diffracted well, and the data reported in the present article are those of the best crystal we were able to find. SCXRD data of 4d were collected on a Bruker Apex-II diffractometer equipped with a CCD detector (T = 100 K, Cu − Kα radiation (λ = 1.54184 Å). Data were collected with the APEX2 software [23], whereas data integration and reduction were performed with the Bruker SAINT software [24]. The crystal structure was solved using the SIR-2004 package [25] and refined by full-matrix least squares against *F^2^* using all data (SHELXL-2018/3) [26]. All the non-hydrogen atoms were refined with anisotropic displacement parameters, whereas all the hydrogen atoms were set in in calculated positions and refined in accordance with the atoms to which they are bonded. 

Geometrical calculations were performed by PARST97 [27], and molecular plots were produced by the program Mercury (v4.1.2) [28] and Discovery Studio Visualizer 2019 [29]. Crystallographic data and refinement parameters are reported in Appendix A. In Appendix A, an Ortep-3 view of the asymmetric unit of 4d is reported. 

### 3.5. Molecular Modeling

The structure of the channel was obtained from the hPANX-1 (PDB 6WBI), with consideration of all the amino acids within a distance of about 25 Å from the center of the pore, which was identified as the ‘CBX binding site’. The ligands were placed at the binding site through AutoDock 4.2 [30]. The main parameters used in this work are in Table 7. With these parameters, AutoDock provides 10 conformations (poses) of the complex ligand interaction site. The conformations are then assembled into clusters within which the structural difference falls within 2.0 RMSD. The program estimates the free energy of binding for each ligand-site interaction complex. The interactions identified by docking are those that have an HBD/HBA distance < 3 Å, and in this case, we can speak of hydrogen bonding interaction; instead, we speak of VdW interactions when the interactions involve aromatic rings and systems capable of such interactions (electrostatic) with a distance < 3 Å.

## 4. Conclusions

Here, we report new Panx-1 quinoline-based inhibitors which reflect the structural requirements for channel blocking of our previously published compounds. Many new products are capable of blocking the Panx1 channel by more than 50% at the screening dose of 50 µM, and the most relevant compounds are the quinolones **6f** and **6g** with an IC_50_ = 3 and 1.5 µM, respectively, which were also found to be selective for Panx-1 (versus P2X7R and Connexin). The quinoline derivative **13e** containing a portion of the drug Probenecid also showed good activity (IC_50_ = 4.2 µM) but binds the purinergic P2X7 receptors. Because the predicted intestinal absorption of the most potent and selective Panx-1 blockers **6f** and **6g** was low, and given that the ester precursors of type **4** resulted inactive, we evaluated the stability of these latter in human plasma with the aim of promoting activation to acid compounds by esterases after absorption; unfortunately, the ester derivatives are highly stable in human plasma samples with t_1/2_ > 120 min. Nevertheless, the most interesting compound **6g** remains an excellent candidate for future in vivo studies on neuropathic pain, but it is also a lead compound for further chemical manipulations. Finally, from molecular modeling studies, it emerged that, similarly to CBX, compounds **6g** and **13e** form strong interactions with Arg75 and Trp74, whereas compound **10** is not able to interact with these amino acids, and this fact could explain its inactivity. 

## Data Availability

Not applicable.

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
