# Peer review of "Design, Synthesis and Pharmacological Evaluation of New Quinoline-Based Panx-1 Channel Blockers"

_ijms, 2023, doi:10.3390/ijms24032022_

Round 1

Reviewer 1 Report

The paper written by Letizia Crocetti et al. reported the design, synthesis and pharmacological evaluation of new quinoline- based Panx-1 blockers. The study is completed by chemical stability X-ray crystallography measurements and molecular modelling calculations. The paper is interesting and with high scientific soundness. I have some observations:

- page 8 line 248-258 the Authors could explain better the results reported in table 3.

- in Figure 7 the molecule 2 (13e) is the only system passively absorbed by the gastrointestinal tract but at the line 321 the Authors indicate the molecule 13g. Is what the Authors have written an error, or do they need to explain more?

- The caption of Figures 11, 12 and 13 (A and B) must be extended.

- The Authors could be better described Figures 11 and 12 and extend the explanation.

- In the conclusions line 512, 13g should be replaced by 13e.

- In the calculations of Figure 13 the compound 10 is present three times, whereas in Figures 11 and 12 compounds 6g and 13e are present as one molecule. How come this different stoichiometry?

- The CHN analysis is indicate but not reported for all compounds.

- Figures 3,4, 5 and 6 reported the quantitative analysis of the % inhibition of Panx-1 for compounds 6f, 6g, 13e and 13gCould the Authors combine the four Figures into one, as shown in Figures S1?

- Figure S1 reported the compounds 6g, 6f, 13e and 13e instead of 13g. Is this a mistake? If it is an error, the Figure S1 must be deleted.

- Why is the stability of KEE in PBS not reported?

- In Table S2 the Authors could be add the type of solvent A and B.

- In Figures S2-S6 the scale on the x and y axes could be adjusted. The experimental data can all be put into one Figure.

Author Response

The paper written by Letizia Crocetti et al. reported the design, synthesis and pharmacological evaluation of new quinoline- based Panx-1 blockers. The study is completed by chemical stability X-ray crystallography measurements and molecular modelling calculations. The paper is interesting and with high scientific soundness. I have some observations:

- page 8 line 248-258 the Authors could explain better the results reported in table 3.

In order to better explain the results reported in table 3, we have changed the sentence: “The IC50 values are reported in Table 3 and the potent Panx-1 blocker 6g, which almost completely inhibited the channel at 50 µM, exhibited interesting efficacy with IC50 = 1.5 µM, lower than most of the reference compounds. Appreciable inhibition values were also obtained for compounds 6f and 13e (IC50 = 3.0  and 4.2 µM respectively). Moreover, compounds 6f, 6g and 13g were selective for Panx-1 channel, while 13e also blocked P2X7 receptors, although with a lower potency (32% versus 88% for Panx).”

with

 “With the exception of 13g, all compounds show IC50 lower than the reference compounds CBX (IC50 = 5 μM) and the potent Panx-1 blocker 6g, which almost completely inhibited the channel at 50 µM, exhibited interesting efficacy with IC50 = 1.5 µM. Appreciable inhibition values were also obtained for compounds 6f and 13e (IC50 = 3.0  and 4.2 µM respectively). Moreover, we tested the selected compounds 6f, 6g, 13e and 13g on oocytes expressing the P2X7 receptor, since many P2X7 receptor ligands are also able to act as Panx-1 channels blockers; at the same time, Connexins have been chosen as additional target because, together with Pannexins and Innexins belongs to the superfamily of “gap junction” proteins. The results clearly indicate that compounds 6f, 6g and 13g were selective for Panx-1 channel, while 13e also blocked P2X7 receptors, although with a lower potency (32% versus 88% for Panx).”

- in Figure 7 the molecule 2 (13e) is the only system passively absorbed by the gastrointestinal tract but at the line 321 the Authors indicate the molecule 13g. Is what the Authors have written an error, or do they need to explain more?

We apologize for the error and we have inserted the right compound

- The caption of Figures 11, 12 and 13 (A and B) must be extended.

Done (now Figures 9-11)

- The Authors could be better described Figures 11 and 12 and extend the explanation.

Done (now Figures 9 e 10)

- In the conclusions line 512, 13g should be replaced by 13e.

 Done

- In the calculations of Figure 13 the compound 10 is present three times, whereas in Figures 11 and 12 compounds 6g and 13e are present as one molecule. How come this different stoichiometry?

Figure 13 (now Figure 11) shows three different poses with low energy of compound 10, all located in the center of the channel. In the case of compounds 6g and 13e, the poses at lower energy are located in a different position of the channel, as shown in figures 11 and 12 (now Figures 9 e 10).

- The CHN analysis is indicate but not reported for all compounds.

We apologize for the oversight, and we have included the table of elemental analysis in the supporting information (Table S12)

- Figures 3,4, 5 and 6 reported the quantitative analysis of the % inhibition of Panx-1 for compounds 6f, 6g, 13e and 13g. Could the Authors combine the four Figures into one, as shown in Figures S1?

Thanks for the suggestion, which makes it clearer to read the data. We have inserted the quantitative analysis of Panx-1% inhibition for compounds 6f, 6g, 13e and 13g into the same figure (now Figure 3) and deleted the single graphic.

- Figure S1 reported the compounds 6g, 6f, 13e and 13e instead of 13g. Is this a mistake? If it is an error, the Figure S1 must be deleted.

Yes, it’s a mistake. We have corrected and replaced 13c with 13g. In any case, Figure S1 was removed from supporting information and inserted in the main manuscript (now Figure 3).

- Why is the stability of KEE in PBS not reported?

KEE is the reference compound (as reported in the cited ref. 20) used only to evaluate the hydrolytic activity of the lot of human plasma used during the stability test. Therefore, its PBS stability was not carried out.

- In Table S2 the Authors could be add the type of solvent A and B.

The authors, following the reviewer’s suggestion, added the composition of solvent A e B in the notes of Table S2.

- In Figures S2-S6 the scale on the x and y axes could be adjusted. The experimental data can all be put into one Figure.

The authors have modified the scale of Figures S2-S6 (now S1-S5) but believe that, by merging the PBS and HPl plots of each compound, the resulted graph could be unclear. In any case, we have inserted both experimental data into one figure for each compound as requested by the Reviewer.

Reviewer 2 Report

The article describes new Panx-1 quinoline based inhibitors. Pannexins are recently discovered channels, and constant advances in biological insight of Panx1 protein are still undergoing, making this target very attractive for Medicinal Chemists.

The group already published 2 research articles describing new Panx-1 inhibitors, demonstrating knowledge and expertizes on this research field.

The article is well written and the logical passages that describe the research plan are well described. I have some suggestions to improve and complete the manuscript in some parts.

Introduction:

-         the type of interaction between the Panx-1 channel and the reported compounds from the literature is not focused. It is not clear if the compounds bind to the same site, or if it was ever determined (CBX was co-crystallized with the target protein, this can at least be mentioned). This would help the reader understand the design of the molecules.

-         Page 2, line 48: please remind the observed in vivo effect.

-         Why the quinolone scaffold was used in these new compound? To bioisosterically replace the indole of structure in panel B? or to mimic the mefloquine? In this case, a parallel between supposed binding mode (see previous comment) would help.

Chemistry:

-         Page 3 line 81: Remove “in-depth”:  the techniques used to assign the structures (13C-NMR, HSQC and HMBC are considered standard techniques).

-         Line 82: which compounds have chemical shift values of 77 ppm and 61 ppm? I saw bidimentional spectra of 4c and 4e, methylene is sround 54 ppm, 5c was 77 ppm.

-         Line 88: the authors reported they observed “the coupling between the methylene protons and the quinolinone C2 carbon for the N-alkylated compounds (4c and 4e),” instead I can see only the coupling between Carbon of methylene and proton of C-2 of quinolinone, for 4c. Moreover, is it possible to detect coupling between methylene protons and C-4 of quinolone in compounds 5c and 5e?

-         Capture of scheme 3: change “amine” with “aniline”

Biological evaluation:

-         To better understand the potency of the described compound, at least the IC50 value of 1 reference compound, tested in the same condition, need to be reported.

-         Figure 3, 4, 5 and 6 should be grouped in a single image.

X-ray crystallography:

-         Why do the authors crystallize compound 4d? This is not a final compound, and no justification is given about this choice. Importantly, in the chemical description of compound 4d, it is reported as an oil. How could the authors crystallize it?

Molecular modeling:

-         In figure 11, an additional box need to be added, showing the experimental binding mode of CBX in binding site, obtained in pdb 6WBI.

-         In figure 11b, 2 residues are labelled as Arg75. Is it a mistake or they belong to different subunits? In this last case, this should emerge from the text or the caption.

Supplementary:

-         Why carbon spectra contain negative peaks?

-         I suggest to report the formula near the proton spectrum for every molecule, it is clearer to read.

Author Response

The article describes new Panx-1 quinoline based inhibitors. Pannexins are recently discovered channels, and constant advances in biological insight of Panx1 protein are still undergoing, making this target very attractive for Medicinal Chemistry. The group already published 2 research articles describing new Panx-1 inhibitors, demonstrating knowledge and expertise on this research field.

The article is well written and the logical passages that describe the research plan are well described. I have some suggestions to improve and complete the manuscript in some parts.

Introduction:

-        The type of interaction between the Panx-1 channel and the reported compounds from the literature is not focused. It is not clear if the compounds bind to the same site, or if it was ever determined (CBX was co-crystallized with the target protein, this can at least be mentioned). This would help the reader understand the design of the molecules.

At present, the only binding site identified is that for CBX [Ruan et al., 2020] through cryoEM technique, while on the other literature compounds we mentioned, no molecular modelling studies or cryoEM techniques have been performed. Therefore there is currently no information on the binding site, let alone whether all Pan-1 blockers bind the same site.

-        Page 2, line 48: please remind the observed in vivo effect.

      We have added: “They showed an anti-hypersensitivity profile and being able to significantly increase the mouse pain threshold 45 min after the intrathecally administration of the doses of 1 nmol and 3 nmol”.   

-         Why the quinolone scaffold was used in these new compounds? To bioisosterically replace the indole of structure in panel B? or to mimic the mefloquine? In this case, a parallel between supposed binding mode (see previous comment) would help.

Regarding the rationale for choosing the quinoline scaffold, we took the inspiration from the indole derivatives shown in Figure 1 panel B, and we formally 'expanded' the nitrogen ring, hoping for a bioisostery effect. This choice was certainly also influenced, as explained in the Introduction, by the fact that for the antimalarial Mefloquine, with quinoline scaffold, a Panx-1-blocking profile was highlighted.

Chemistry:

-        Page 3 line 81: Remove “in-depth”:  the techniques used to assign the structures (13C-NMR, HSQC and HMBC are considered standard techniques).

      Done

-        Line 82: which compounds have chemical shift values of 77 ppm and 61 ppm? I saw bidimensional spectra of 4c and 4e, methylene is around 54 ppm, 5c was 77 ppm.

      We apologize for the error, we have replaced 61 ppm with 54 ppm, which is the correct ppm value

-        Line 88: the authors reported they observed “the coupling between the methylene protons and the quinolinone C2 carbon for the N-alkylated compounds (4c and 4e),” instead I can see only the coupling between Carbon of methylene and proton of C-2 of quinolinone, for 4c. Moreover, is it possible to detect coupling between methylene protons and C-4 of quinolone in compounds 5c and 5e?

      You are right. While in the HMBC spectrum of 4e we can see the coupling between C(2)H and C of methylene (signal at 54.7) and the coupling between protons of methylene and C2 (signal at 150.2), in the spectrum of 4c only one coupling is seen, precisely signal between C(2)and C of methylene group of the chain (signal at 54.1), probably due to a less resolution of the spectrum.

In compounds 5c and 5e the signal at about 164 ppm indicate the coupling between the methylene protons of OCH2 and C4, confirming the correct assignment of the two structures.

-        Capture of scheme 3: change “amine” with “aniline”

      Done

Biological evaluation:

-         To better understand the potency of the described compound, at least the IC50 value of 1 reference compound, tested in the same condition, need to be reported.

      Thanks for the suggestion. We have added the IC50 of CBX (reference compound) in table 3 with the corresponding reference.

-         Figure 3, 4, 5 and 6 should be grouped in a single image.

      Done (see Figure 3 in the main text).

X-ray crystallography:

-         Why do the authors crystallize compound 4d? This is not a final compound, and no justification is given about this choice. Importantly, in the chemical description of compound 4d, it is reported as an oil. How could the authors crystallize it?

      Since the alkylation of intermediate 3 was the first step in the synthetic procedure, highlighting the tautomerism of the quinolone nucleus and thus providing the two isomers, we consider it was important at this level to understand the exact structure of the two compounds formed, so as to carry out the hydrolysis of only the ester derivative of interest.

      Also, we apologize for the error related to compound 4d, which is not an oil, and we have included the melting point.

Molecular modeling:

-         In figure 11, an additional box need to be added, showing the experimental binding mode of CBX in binding site, obtained in pdb 6WBI.

      We have inserted Figure 8 in the main text about the binding mode of CBX.

-         In figure 11b, 2 residues are labelled as Arg75. Is it a mistake or they belong to different subunits? In this last case, this should emerge from the text or the caption.

      We have changed the caption of Figure 11 (now Figure 9)

Supplementary:

-        Why carbon spectra contain negative peaks?

We have performed a 13C-APT NMR. It’s a useful way and a simple method to assign C-H multiplicities in 13C NMR spectra. It provides information on all carbon multiplicities within a single experiment.

The APT (or J-resolved) experiment yields methine (CH) and methyl (CH3) signals negative and quaternary (C) and methylene (CH2) signals positive. 

-         I suggest to report the formula near the proton spectrum for every molecule, it is clearer to read.

      Thanks for the suggestion. We reported the structure of compounds in the proton spectrum.

Round 2

Reviewer 2 Report

The paper has been improved taking into considerations the suggestion of the referees.